# Implications of RAS Mutations on Oncological Outcomes of Surgical Resection and Thermal Ablation Techniques in the Treatment of Colorectal Liver Metastases

**DOI:** 10.3390/cancers14030816

**Published:** 2022-02-05

**Authors:** Rami Rhaiem, Linda Rached, Ahmad Tashkandi, Olivier Bouché, Reza Kianmanesh

**Affiliations:** 1Faculty of Medecine, University Reims Champagne-Ardenne, 51100 Reims, France; obouche@chu-reims.fr (O.B.); rkianmanesh@chu-reims.fr (R.K.); 2Hepatobiliary, Pancreas, Endocrine and Digestive Surgical Oncology Department, Robert Debré Hospital, CHU de Reims, 51100 Reims, France; lrached@chu-reims.fr (L.R.); atashkandi@chu-reims.fr (A.T.); 3Digestive Oncology and Hepatogastroenterology Department, Robert Debré Hospital, CHU de Reims, 51100 Reims, France

**Keywords:** RAS mutations, colorectal cancer, liver metastases, surgery, resection, ablation

## Abstract

**Simple Summary:**

Modern management of colorectal liver metastases (CRLM) requires a thorough knowledge of tumor biology and oncogenes mutations. RAS mutations are of paramount interest for the indication of targeted therapies and is increasingly considered as a negative prognostic factor for patients undergoing surgical resection or ablation for CRLM. Several studies discussed the results of specific technical considerations according to RAS mutational status on the oncological outcomes after surgical resection/ablation for CRLM. We reviewed the available data on the real impact of RAS mutations on the prognosis with special regard to the need of a tailored surgical (ablation) approach according to tumoral biology.

**Abstract:**

Colorectal cancer (CRC) is the third most common cancer worldwide and the second leading cause of cancer-related death. More than 50% of patients with CRC will develop liver metastases (CRLM) during their disease. In the era of precision surgery for CRLM, several advances have been made in the multimodal management of this disease. Surgical treatment, combined with a modern chemotherapy regimen and targeted therapies, is the only potential curative treatment. Unfortunately, 70% of patients treated for CRLM experience recurrence. RAS mutations are associated with worse overall and recurrence-free survival. Other mutations such as BRAF, associated RAS /TP53 and APC/PIK3CA mutations are important genetic markers to evaluate tumor biology. Somatic mutations are of paramount interest for tailoring preoperative treatment, defining a surgical resection strategy and the indication for ablation techniques. Herein, the most relevant studies dealing with RAS mutations and the management of CRLM were reviewed. Controversies about the implication of this mutation in surgical and ablative treatments were also discussed.

## 1. Introduction

Management of colorectal liver metastases (CRLM) has evolved considerably during recent decades. Indeed, the use of a combination of perioperative medical therapy and surgical treatment remains the standard of care. The tremendous progresses of chemo- and targeted therapies regimens have been achieved allowing a higher rate of conversion (15–30%), from initially unresectable to resectable CRLM [1,2,3]. Additionally, recent advances of surgical techniques, including portal vein embolization and total venous deprivation to prepare 2-stage hepatectomy, associating liver partition and portal vein ligation for staged hepatectomy (ALPPS) and ablative treatments, allowed a wider indication of surgery [4]. This explains the high survival rates after surgery in selected patients for CRLM up to 40–65% at 5-year [2,5], and 25% at 10-year [5], while such long survival rates are uncommon after chemotherapy alone.

Tumor biology, in particular RAS mutations, is obviously among the strongest prognostic factors of CRLM. It is of paramount interest in the choice of the appropriate chemo- and targeted therapy regimens. Several recent studies suggested a clear implication of tumor biology in defining the optimal surgical/ablation techniques and margins.

In this review, we will report and discuss data reporting the role of RAS mutations in tailoring the surgical and/or ablation approach.

## 2. RAS Mutations and Prognosis after CRLM Treatment

The rat sarcoma viral oncogenes (RAS) family (KRAS, NRAS and HRAS) plays a pivotal role in the promotion of tumoral cell growth, angiogenesis and the invasiveness of the tumor through the mitogen-activated protein kinase (MAPK) signaling pathway [6]. This latter is continuously activated in case of a RAS mutation, resulting in resistance to anti-EGFR therapies [7].

Interestingly, several studies have reported a prognostic impact of a RAS mutation in patients undergoing CRLM resection. The Memorial Sloan–Kettering Cancer Center (MSKCC) group [8] were the first to report that the KRAS mutation was associated with worse disease specific survival than the KRAS wild type after both primary (median 2.6 vs. 4.8 years; *p* = 0.0003) and liver metastases resection (median 2.7 vs. 6 years; *p* = 0.004). The presence of an additional high Ki-67 expression harbored even worst survival rates. 

Since then, incidences of the KRAS mutation reported in the surgical series ranged from 15 to 50%, with a shorter overall survival (OS) and recurrence-free survival in many studies (Table 1). The MD Anderson group reported a significant association between the RAS mutation and both the shortness of the time interval to recurrence and the rate of recurrence above all local treatments [9].

Goffredo et al. [23] explored the prognostic factors in a large cohort of 2655 patients enrolled from the US National Cancer Database. All patients were treated, between 2010 and 2015, for synchronous CRLM with concomitant resection of the primary tumor and metastases. The KRAS mutation and right-sided primary tumor were among the major prognostic factors associated with worse OS [23]. NRAS mutations, more infrequently observed, were also correlated to unfavorable oncological outcomes [29,30]. The RAS mutation was integrated in two recent clinical risk scores predicting survival after CRLM resection: the genetic and morphological evaluation “GAME score” [31,32], and the “modified-Clinical risk score” (m-CRS). These scores achieved better discriminatory power than the “Fong’s Clinical risk score” [33]. The MD Anderson group has recently published the “Contour prognostic model” that was designed following the concept of the “Metroticket score”, previously developed to predict survival after liver transplantation for hepatocellular carcinoma beyond the Milan criteria [34]. This score was validated by an international multicentric cohort. It is based on the diameter and number of lesions considered as continuous variables along with the RAS mutation status. It showed a good prediction power for OS after the resection of CRLM [35]. However, more recently, Tsilimigras et al. [36] reported a poor prediction power of the “Tumor Burden Score” in KRAS mutated tumors. The Tumor Burden score reflected the morphologic characteristics of metastases based on the maximum tumor size and number of lesions [37]. The authors reviewed, in an international multi-institutional database, the results of 1361 patients who underwent hepatic resection for CRLM and analyzed the prognostic impact of the Tumor Burden Score depending on the KRAS status. This score was associated with worse overall survival for the KRAS wild type but not for KRAS mutated tumors [36].

Although there is a large consensus on the negative prognostic impact of RAS mutations after liver surgery for CRLM, several recent data suggested an overestimation of its value, in particular, the possibility of different biological patterns between RAS mutants with there, subsequently, being a difference in their effect on the risk of recurrence and survival after treatment of CLRM [38,39]. Xie et al. [40] reported, in a cohort of 323 patients treated for CRLM, that the prognostic impact of the KRAS mutational status was more significant when the primary tumor was left-sided. Sakai et al. [27] analyzed the results of 101 patients, among them 38 patients with the KRAS mutation, and concluded that the KRAS mutation was an independent prognostic factor only for synchronous CRLM. Several investigators assessed the impact of the mutation location on the prognosis. Frankel et al. [38] showed that NRAS and KRAS mutations were present in 43% of patients, the majority being KRAS mutations (number of KRAS mutations = 65, number of NRAS mutations = 6). The location of the mutation was in exon 2 (codon 12 or 13) in 81.6%, exon 3 in 10% and exon 4 in 8.5% of RAS mutations. According to the location of the mutation, patients exhibited various tumoral features. Indeed, the exon 2 mutation resulted in similar features as the RAS wild type, with a median size of nodules < 5 cm and an average of 2.4 tumors per resection. The exon 3 mutation seemed to be associated with multiple but smaller nodules that tend to occur early after the primary tumor resection, whereas patients with the RAS mutation in exon 4 had solitary CRLM but were larger in size, and had a longer time interval after the resection of the primary tumor than the exon 3 mutation. Authors from the same group recently actualized their data with 938 patients treated for CRLM with sufficient tumor genomic profiling. The KRAS mutation was present in 47% of patients with 91.5% of mutations in exon 2, 3.1% in exon 3, and 5.4% in exon 4. The NRAS mutation was found in only 4.2% of patients with mostly mutations in exons 2 and 3 (53% and 41.2%, respectively). K/NRAS mutations were associated with worse OS with a tendency towards more favorable oncological results in patients with the exon 4 mutation. In the same setting, Margonis et al. [18] reported a worse prognosis when the KRAS mutation was in codon 12 when compared to it in codon 13. Among all mutations of codon 12, only patients with G12S and G12V mutations seemed to have a worse oncological outcome than KRAS wild-type patients. Meanwhile, in another study of the John Hopkins Group, KRAS codon 13 mutations seemed to be associated to a higher risk of extrahepatic recurrence than codon 12 mutations, especially in the pulmonary location [41].

More importantly, other oncogenes are valuable aside the RAS mutational status to predict optimally the prognosis after CRLM resection. Indeed, as shown by Kawaguchi and the MD Anderson group, the association of RAS, TP53 and/or SMAD4 seems to be accurately correlated to worse OS and recurrence free survival (RFS) in 507 patients undergoing surgical resection for CRLM [42]. Furthermore, the authors found no difference in OS and RFS between RAS mutated with wild-type TP53—SMAD 4 and RAS wild-type patients [42].

These data suggested differences in the tumoral pattern and in oncological outcomes according to the location (exon and codon) of the mutation and to the associated mutations.

## 3. Implication of RAS Mutations in the Surgical Resection of CRLM

Modern surgical management of CRLM is based on the concept of “parenchymal-sparing” surgery, shifting the paradigm from anatomical and large resections to limited resections with a surgical margin ≥ 1 mm, resulting in comparable survival outcomes with lower postoperative morbidity and mortality rates [42,43,44,45,46,47]. Moreover, R1 vascular is an acceptable surgical option in case of direct contact between the nodule and major vascular structures [48,49]. Contrariwise, a positive resection margin (R1 parenchymal) is associated with a higher rate of local recurrence and worse prognosis. In this context, the surgical margin for RAS mutated CRLM is a matter of debate. Brudvik et al. [50] have reported an association between the RAS mutation and the depth of the resection margins in patients undergoing liver resection for CRLM (hazard ratio (HR) = 2.439; *p* = 0.005). Patients with liver-first recurrence of RAS-mutated CRLM had significantly narrower margins than patients with RAS wild type tumors (4 mm vs. 7 mm; *p* = 0.031) [50]. The same conclusions were recently reported by Zhang et al. [51] in a consecutive cohort of 251 patients treated for CRLM with more micrometastases, thicker margins and a higher rate of R1 resection in the KRAS mutated group [51]. To overcome this problem, Margonis et al. [41] suggested a significant benefit from anatomical resection in KRAS-mutated CRLM, as it seems to allow better liver-specific disease-free survival (DFS) than non-anatomical resections in a multicentric cohort of 389 patients with 140 patients (36%) presenting with KRAS-mutated CRLM (33.8 vs. 10.5 months; *p* < 0.001). Such a difference was not observed in the KRAS-wild type group. The main flaw of this study was a higher rate of ablation procedures in the non-anatomical group (32% vs. 8%) and the absence of analysis of the sub-group of patients treated only with liver resection. This point might alter the interpretation of the survival difference in favor of anatomical resections [52]. Meanwhile, recently, Joechle et al. [53] found no significant difference in OS and RFS between anatomical and non-anatomical resection in 622 patients treated for CRLM with a documented RAS mutation status before and after propensity score matching. In view of these results, the MD Anderson group recommends, when anatomically feasible, wider planned resection margins (≥15 mm) in the case of RAS mutated-CRLM [52], which is debatable. Conversely, the John Hopkins hospital group analyzed the impact of surgical margins after resection of CRLM according to the RAS mutation status [54]. Margonis et al. [54] compared the outcomes after the R0 and R1 resection, and subsequently subdivided the R0 resection group into 3 subgroups according to the width of the surgical margins: 1–4, 5–9 and ≥10 mm. In the KRAS wild type group, the R1 resection was associated with worse OS compared to the R0 resection, but wider margins did not confer an additional OS benefit. In the other hand, for the KRAS mutation group, the OS of the R0 resection, regardless of the width of margin, was not better than the R1 resection group. The same conclusions were drawn in a more recent study with 500 patients [26]. While the resection margin seemed to be associated to death-censored liver-specific recurrence-free survival, it did not impact survival outcomes for KRAS mutated patients [26]. These results stressed the importance of tumor biology and aggressiveness of RAS-mutated CRLM that outbalance the prognostic impact of the surgical margin width. Furthermore, R1 vascular resection seems to harbor a lower risk of local recurrence in KRAS mutated CRLM [55]. The Humanitas group evaluated the local recurrence after CRLM resection according to the quality of resection and to the KRAS mutation status [55]. KRAS mutation was not associated to a higher risk of local recurrence in R0 patients. R1 parenchymal resection, exposing the tumor edge during parenchymal dissection, was correlated to a higher rate of local recurrence in mutated KRAS tumors when compared to the KRAS wild type (respectively, local recurrence rate per patient: 25.4% vs. 18.3%; *p* = 0.404, in situ local recurrence rate: 19.5% vs. 9.9%; *p* = 0.048). Interestingly, results were different for R1 vascular resections (resections with the detachment of nodules from vascular structures). In this regard, the local recurrence rate was higher in the KRAS wild type subgroup (local recurrence rate per-patient 14.6% vs. 2%, *p* = 0.043, in situ local recurrence rate 13.3% vs. 1.9%, *p* = 0.046) [55]. These data are valuable and introduced the concept of a tailored surgical approach according to tumor biology in patients treated for CRLM.

## 4. Implication of RAS Mutations in Ablative Treatment of CRLM

Ablation is a valuable treatment of CRLM< 3 cm. This debate around the impact of the RAS mutational status on the oncological outcomes of surgical resection also brought the same questioning. Obviously, all published studies reported shorter local tumor progression-free survival in RAS mutated [56,57] and KRAS mutated patients with CRLM [58,59]. Even if all these studies included patients with tumors larger than 3 cm, which is questionable, in multivariate analysis, the 2 main risk factors of local tumor progression were mutational status and ablation margins (Table 2 and Table 3). These data were also confirmed by a more recent study from the Amsterdam group [60]. The authors analyzed the impact of primary tumor sidedness, genetic mutations (RAS and BRAF) and the microsatellite instability status to determine the prognosis of patients treated for CRLM enrolled in the Amsterdam Colorectal Liver Met Registry. RAS mutation was associated to shorter local tumor progression-free survival and to lower local control rates after thermal ablation. In these studies, the optimal minimal ablation margin was >5 mm [56,58] and raised to 10 mm [57,59]. Wider margins seem to be necessary to reduce rates of local tumor progression in RAS/KRAS mutation patients [60].

## 5. Conclusions

RAS mutations seem to present a negative impact on the oncological outcome of patients treated for CRLM. Several studies pointed to the importance of a multidisciplinary “tailored” approach of CRLM according to the RAS mutational status to choose the optimal preoperative treatment and to optimize the surgical resection and/or ablation technique and planned margins. However, larger studies with genetic sequencing are required to assess a more thorough analysis of the real impact of the RAS mutational status according to the exon/codon location of the mutations, and, more specifically, to the association of other somatic mutations such BRAF, TP 53, PIK3CA and/or SMAD 4 that may harbor a poorer prognostic outcome. Interestingly the primary tumor side and RAS mutation might also be considered. Such a tailored approach, considering the whole genetic profiling of the tumor, will allow further advancement in the knowledge of tumor biology and may be valuable for the management and counseling of patients treated for metastatic colorectal cancer.

## Figures and Tables

**Table 1 cancers-14-00816-t001:** Summary of studies reporting survival outcomes of treatment of colorectal liver metastases according to RAS/KRAS mutations.

Study	N *	RAS/KRAS Mutation (%)	Overall Survival (OS)		Recurrence/Disease Free Survival (RFS/DFS)	
			Clinical Parameter	HR (95% CI); *p*-Value	Clinical Parameter	HR (95% CI); *p*-Value
Petrowsky et al., 2001 [10]	41	6 (15%)	Survival	1.39 (0.45–4.27); *p* = 0.57	N/A	N/A
Nash et al., 2010 [8]	188	51 (27%)	5-year survival	2.4 (1.4–4.0); *p* = 0.001	N/A	N/A
Teng et al., 2012 [11]	292	111 (38%)	Median OS	1.48 (0.86–2.56); *p* = 0.156	N/A	N/A
Stremitzer et al., 2012 [12]	76	15(20%)	5-year survival	3.51 (1.30–9.45); *p* = 0.013	3-year RFS	2.48 (1.26–4.89); *p* = 0.009
Karagkounis et al., 2013 [13]	202	58 (29%)	3-Year OS	1.99 (1.21–3.26); *p* = 0.007	3-Year RFS	1.68 (1.04–2.70); *p* = 0.034
Isella et al., 2013 [14]	64	21 (33%)	N/A	N/A	Median DFS	1.58 (0.79–3.16); *p* = 0.19
Vauthey et al., 2013 [15]	193	27 (14%)	3-year OS	2.26 (1.13–4.51); *p* = 0.002	3-year RFS	1.92 (1.21–3.03); *p* = 0.005
Kemeny et al., 2014 [16]	169	51 (30.2%)	3-year OS	2.0 (0.87–4.46); *p* = 0.104	3-year RFS	1.9 (1.16–3.31); *p* = 0.01
Shoji et al., 2014 [17]	108	39 (36.1%)	N/A	N/A	Median RFS	1.91 (1.163–3.123); *p* = 0.01
Margonis et al., 2015 [18]	331	91 (27.5%)	Median OS	1.7 (1.13–2.55); *p* = 0.01	Median/5-year RFS	*p* = 0.57
			Codon 12 mutant Codon 13 mutant	1.61 (0.87–2.97); *p* = 0.13		
Sasaki et al., 2016 [19]	129	78 (48.8%)	Median/5-year OS	1.37 (0.98–1.91); *p* = 0.06	Median/5-year RFS	1.10 (0.85–1.44); *p* = 0.47
	297	68 (28.8%)				
Shindoh et al., 2016 [20]	163	74 (45%)	3-Year OS Disease specific survival	2.86 (1.36–6.04); *p* = 0.006	3-Year RFS	1.47 (1.00–2.15);
					Liver RFS	*p* < 0.048
						3.5 (2.14–5.73); *p* < 0.001
Amikura et al., 2018 [21]	421	191 (43.8%)	5-Year OS	1.67 (1.19–2.38); *p* = 0.0031	5-year RFS	1.70 (1.206–2.422); *p* = 0.0024
O’Connor et al., 2018 [22]	662	174 (26.3%)	Death	1.11 (0.73–1.69); *p* = 0.207	Recurrence	1.42 (1.10–1.85); *p* = 0.008
Goffredo et al., 2019 [23]	2655	1116 (42%)	5-Year OS	1.21 (1.04–1.39); *p* = 0.012	N/A	N/A
Brunsell et al., 2020 [24]	106	53 (50%)	3-year CSS (cancer specific survival)	3.3 (1.6–6.5); *p* = 0.001	N/A	N/A
Kim et al., 2020 [25]	227	78 (34%)	Median OS	1.420 (0.902.25); *p* = 0.042	Median RFS	1.137 (0.83–1.55); *p* < 0.001
Hatta et al., 2021 [26]	500	152 (30.4%)	5-year OS	1.52 (1.14–2.03); *p* = 0.004	5-Year RFS	1.29 (1.00–1.67); *p* = 0.049
Sakai et al., 2021 [27]	101	38 (37.6%)	5-year OS	2.41 (1.36–4.25); *p* = 0.003	3-year RFS	N/A
Saadat et al., 2021 [28]	938	445 (47%)	Median OS	HR 1.67 (1.39–2); *p* < 0.001	Median RFS	1.74 (1.45–2.09); *p* < 0.001

N *: Number of patients included in the study.

**Table 2 cancers-14-00816-t002:** Results of studies reporting implications of RAS mutations on surgical resection of colorectal liver metastases.

Authors	Study Period	N * (%) of RAS/KRAS Mutation	Associated Ablation Procedures	Study Keypoint	Findings	Results
Brudvik et al. [50]	2005–2013	RAS 229/633	N/A	Resection margin	RAS mutation associated:	HR: 2.439; *p* = 0.005
		(36.2%)			- to positive resection margin (<1 mm)	
					-worst OS	HR 1.629; *p* = 0.044
Zhang et al. [51]	2010–2017	KRAS 121/251	N/A	Micrometastasis	KRAS mutation associated with higher rate	KRAS mut vs. KRAS wild 60.3% vs. 40.8%;
		(48.2%)				*p* = 0.002
					higher number and	(median 2.0 (range 0–38.0) vs. median 0 (range: 0–15.0); *p* = 0.001)
					density of micrometastases	56% vs. 43%; *p* = 0.013
				Resection margin	Higher rate of R1 resection (tumoral cell on the resection margin)	21.5 vs. 9.2%; *p* = 0.007
					Narrower resection margin in KRAS mut	median 2.00 (range 0–40.00) vs. 4.30 (range 0–50.00) mm;
						*p* = 0.002
				LRFS	KRAS mut associated with worst LRFS	HR: 1.495 (95% CI: 1.069–2.092); *p* = 0.019
				OS	KRAS mut associated with worst OS	HR: 2.039 (95% CI: 1.217–3.417); *p* = 0.007
Margonis et al. [41]	2000–2015	KRAS 140/389 (36%)	NAR:53/165 (32%)	Anatomical vs. non anatomical resection	AR was associated with better DFS in KRAS mut but not in KRAS wild	DFS:
						KRAS mut HR: 0.45 (95%
			AR:19/224 (8.5%)			CI: 0.27–0.74; *p* = 0.002
						KRAS wild: NS
Joechle et al. [53]	2006–2016	RAS 274/622 (40%)	N/A	Anatomical vs. non anatomical resection	No difference in OS and Live specific RFS before and after PSM	
					RFS was better in the AR before PSM but not after PSM	
Margonis et al. [54]	2003–2015	KRAS 153/411(37.2%)	84 (20.4%)	Impact of resection margin width on OS according to KRAS status	KRAS wild type: R0 resection was associated to better OS than R1 resection (<1 mm) with no benefit from wider margin (1–4 mm; 5–9 mm;>9 mm)	KRAS wild:
					KRAS mut: No difference in OS between R0 and R1 resection, regardless of the width of surgical margin	R1 ref
						1–4 mm: HR: 0.45, 95%CI: 0.24–0.85; *p* = 0.014)
						5–9 mm: HR: 0.35, 95%CI: 0.17–0.70; *p* = 0.003
						>9 mm: HR: 0.33, 95%CI: 0.16–0.68; *p* = 0.002
						KRAS mut:
						1–4 mm: HR: 0.80, 95%CI: 0.38–1.70; *p* = 0.522
						5–9 mm: HR: 0.68, 95%CI: 0.30–1.54; *p* = 0.356
						>9 mm: HR: 1.08, 95%CI: 0.50–2.35; *p* = 0.844
Hatta et al. [26]	2011–2016	KRAS 152/500 (30.4%)	N/A	Impact of resection margin width on OS, RFS and LS-RFS according to KRAS status	KRAS wild type:	
					Resection margin width was associated to a better OS, RFS (Death censored) and LS-RFS (Death censored)	
					KRAS mut:	
					No difference between R0 (regardless to the width of margin) and R1 in all studied survival parameters	
Procopio et al. [55]	2008–2016	KRAS	N/A	Impact of R1 parenchymal and R1 vascular resections on risk of local recurrence after resection according to KRAS status	Higher rates of recurrence in KRas mut after R1 parenchymal resection	R1 parenchymal resection
						(KRAS mut vs. KRAS wild)
						local recurrence rate per patient: 25.4% vs. 18.3%; *p* = 0.404
						in situ local recurrence rate: 19.5% vs. 9.9%; *p* = 0.048
						R1 vascular resection
						(KRAS mut vs. KRAS wild)
						local recurrence rate per patient 2% vs. 14.6%; *p* = 0.043,
						in situ local recurrence rate
		155/340 (46%)			Higher rates of recurrence in KRAS wild after R1 vascular resection	1.9%, vs. 13.3%; *p* = 0.046

N */(%): Number and percentage of RAS/KRAS mutations in the study, N/A: Not mentioned, HR Hazard ratio, *p*: *p*-value, OS: Overall survival, RFS: Recurrence-free survival, LS-RFS: Liver specific Recurrence-free survival, AR: Anatomical resection, NAR: Non anatomical resection.

**Table 3 cancers-14-00816-t003:** Results of studies reporting ablative treatment for colorectal liver metastases according to RAS mutations.

Study Year	N	Median Size of CRLM	N * of KRAS Mutation	Procedures of Ablation	OS	LTPFS	LC (Site Specific Recurrence)
					HR (95%CI);	HR (95%CI);	HR (95%CI); *p*-Value
					*p*-Value	*p*-Value	
		(cm)			% at 3 Years, *p*-Value		% at 3 Years, *p*-Value
Shady 2017 [58]	97	1.7 (0.6–5)	38/97	Percutaneous RFA	2.0 (1.2–3.3);	1.7 (0.89–3.2)	2.0 (1.0–3.7)
			(exon 2)		*p* = 0.009	*p* = 0.11	*p* = 0.037
Odisio 2017 [56]	92	1.6 (0.4–4.0)	36/92	Percutaneous RFA + MWA	N/A	3.01 (1.60–5.77)	N/A
					40% vs. 82%; *p* = 0.013	*p* = 0.001	56% vs. 43%
							*p* = 0.013
Calandri 2018 [57]	136	1.6 (0.5–5.2)	54/136	Percutaneous RFA, MWA, cryotherapy	N/A	2.85 (1.7–4.6)	N/A
						*p* < 0.001	
Jiang 2019 [59]	76	2.3 (0.9–0.7)	38/76	Percutaneous RFA	Not significant *p * = 0.228	3.24 (1.41–7.41) *p* = 0.005	Not significant
							*p* = 0.356
Dijkstra 2021 [60]	79		36/79	Percutaneous RFA + MWA	N/A	Significantly lower *p* = 0.037	N/A

N * = Number of patients with KRAS mutation, CRLM: Colorectal liver metastasis, OS: Overall survival, HR: Hazard ratio, CI: confidence interval, LTPFS: Liver tumor progression free survival, RFA: Radiofrequency ablation, MWA: Microwave ablation.

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
