# Peer review of "Implications of RAS Mutations on Oncological Outcomes of Surgical Resection and Thermal Ablation Techniques in the Treatment of Colorectal Liver Metastases"

_cancers, 2022, doi:10.3390/cancers14030816_

Round 1

Reviewer 1 Report

please check for spelling errors,

 Lane 6: Faculty…

Lane 7: Surgical Department

Lane 106: please specify better nKRAS, nNRAS (n=number, has to be clear)

Lane 117: 4.2% instead of 4,2%, with the majority of mutations instead …mostly mutations

Lane 119: In the same setting … there is a double space previously…

Lanes 153-154: I suggest 33.8 vs 10.5 months, in this order

Lane 158: Meanwhile, recently….

Lanes 180-182: was correlated to a higher rate of local recurrence in mutated KRAS tumors when compared to KRAS wild type (local recurrence rate per-patient 18.3% vs 25.4%, p = 0.404, in situ local recurrence rate 9.9% vs 19.5%, p = 0.048) – since it is written a higher rate, in mutKRAS vs wtKRAS, the numbers are correctly presented or should be reverse???, please specify!

Tables 1, 2: Addition of explanations of abbreviations under the tables is needed.

Table 2: the bibliographic citations should be noted here as well as in table 1

Tables 1,2: I suggest less columns, year/reference could be included in the study-column, numbers, p-values should also be presented correctly!!!

QUESTIONS

  1. Except KRAS mutation status differences (single mutations), have been described any differences in patients harboring both KRAS-NRAS mutations?

  1. Have been recorded differences between KRAS plus other associated mutations versus NRAS plus other associated mutations???

Author Response

Reviewer 1:

please check for spelling errors,

 Lane 6: Faculty…

Lane 7: Surgical Department

Lane 106: please specify better nKRAS, nNRAS (n=number, has to be clear)

Lane 117: 4.2% instead of 4,2%, with the majority of mutations instead …mostly mutations

Lane 119: In the same setting … there is a double space previously…

Lanes 153-154: I suggest 33.8 vs 10.5 months, in this order

Lane 158: Meanwhile, recently….

Lanes 180-182: was correlated to a higher rate of local recurrence in mutated KRAS tumors when compared to KRAS wild type (local recurrence rate per-patient 18.3% vs 25.4%, p = 0.404, in situ local recurrence rate 9.9% vs 19.5%, p = 0.048) – since it is written a higher rate, in mutKRAS vs wtKRAS, the numbers are correctly presented or should be reverse???, please specify!

Response: We would like to thank the reviewer for these comments. All spelling and grammar errors were corrected.

Tables 1, 2: Addition of explanations of abbreviations under the tables is needed.

Response: Abbreviations were detailed in the bottom of both tables

Table 2: the bibliographic citations should be noted here as well as in table 1

Response: Citations were added in table 2 (changed in table 3)

Tables 1,2: I suggest less columns, year/reference could be included in the study-column, numbers, p-values should also be presented correctly!!!

Response: Years of publication and references were added to the study column to reduce the number of columns in tables 1 and 2. In addition, all p-values were presented in italic after a “;” to present more clearly the results of the different studies. We hope that these modifications will render tables clearer and suitable for publication

QUESTIONS

  1. Except KRAS mutation status differences (single mutations), have been described any differences in patients harboring both KRAS-NRAS mutations?

Response: Mutations of KRAS and NRAS are mutually exclusive. However, several case reports and papers reported the possibility of either spontaneous mutations or de novo mutations during treatment leading tumoral tissue to harbor both mutations. This double mutation is thus rare and currently to our knowledge there is no available data reporting the prognostic impact of KRAS-NRAS mutations for patients treated for CRLM.

  1. Have been recorded differences between KRAS plus other associated mutations versus NRAS plus other associated mutations???

Response: To our knowledge there is no available data comparing the prognostic impact of KRAS and NRAS mutations (with or without associated other mutations) for metastatic colorectal cancer. The study of El Agy et al. (Mutation status and prognostic value of KRAS and NRAS mutations in Moroccan colon cancer patients: A first report. PLoS One. 2021 Mar 30;16(3):e0248522.doi: 10.1371/journal.pone.0248522) reported the worst prognosis with KRAS mutation than in NRAS mutation for colon cancer.

Reviewer 2 Report

The authors reviewed the papers regarding RAS mutations and colorectal liver metastases focusing on the prognosis. Actually, the review provide no novel or important information to the readers of the journal. I am sorry but it is difficult for this paper to be considered further for its publication as a review in this journal.

Author Response

We thank the reviewer for his evaluation of our manuscript. It is true that several previous studies and reviews about RAS mutations and colorectal liver metastases were published. But our manuscript summarizes the most relevant studies dealing with prognostic impact of RAS mutations and surgical/ablation strategies. Also, it gives an update with the most recent surgical studies that brought the reflection of the necessity of further investigations as it seems clearly that KRAS mutated CRLM are heterogenous, that wider resection margin for KRAS mutated patients might not be necessarily the solution to improve the prognosis of patients and further studies exploring the impact of additional mutations are required to improve our knowledge to fashion a “tailored approach” according to tumor biology.

Reviewer 3 Report

The authors reviewed the most relevant studies on the impact of RAS mutations on the oncological outcomes of surgical resection and thermal ablation in the treatment of colorectal liver metastasis (CRLM).

This mini-review is well written, selecting the most recent literature.

It is felt that the 3rd section “Implication of RAS mutations in the surgical resection of CRLM”, a summarized table of the available published data may be helpful for better understanding of the readers.

Author Response

The authors reviewed the most relevant studies on the impact of RAS mutations on the oncological outcomes of surgical resection and thermal ablation in the treatment of colorectal liver metastasis (CRLM).

This mini-review is well written, selecting the most recent literature.

It is felt that the 3rd section “Implication of RAS mutations in the surgical resection of CRLM”, a summarized table of the available published data may be helpful for better understanding of the readers.

Response : We woud like to thank the reviewer for his valuable comments. An additional table (new Table 2) detailing results of studies dealing with surgical data

Round 2

Reviewer 2 Report

We have still concerns to publish this information as a review as once published it will influence the therapeutic decision of the various physicians.